# Relationship between Serum Kallistatin and Afamin and Anthropometric Factors Associated with Obesity and of Being Overweight in Patients after Myocardial Infarction and without Myocardial Infarction

**DOI:** 10.3390/jcm10245792

**Published:** 2021-12-10

**Authors:** Grzegorz Józef Nowicki, Barbara Ślusarska, Maciej Polak, Katarzyna Naylor, Tomasz Kocki

**Affiliations:** 1Department of Family and Geriatric Nursing, Medical University of Lublin, Staszica 6 Str., 20-081 Lublin, Poland; basiaslusarska@gmail.com; 2Department of Epidemiology and Population Studies, Jagiellonian University Medical College, Grzegórzecka 20 Str., 31-531 Krakow, Poland; maciej.1.polak@uj.edu.pl; 3Independent Laboratory of Emergency Medicine, Centre for Medical Simulation, Medical University of Lublin, Chodźki 4 Str., 20-093 Lublin, Poland; Katarzyna.Naylor@hotmail.com; 4Chair and Department of Experimental and Clinical Pharmacology, Medical University of Lublin, Jaczewskiego 8b Str., 20-090 Lublin, Poland; tomasz.kocki@umlub.pl

**Keywords:** kallistatin, afamin, cardiovascular diseases, overweight, obesity

## Abstract

Extensive clinical and epidemiological evidence has linked obesity to a broad spectrum of cardiovascular disease (CVD), including coronary disease, heart failure, hypertension, cerebrovascular disease, atrial fibrillation, ventricular arrhythmias, and sudden death. In addition, increasing knowledge of regulatory peptides has allowed an assessment of their role in various non-communicable diseases, including CVD. The study assessed the concentration of kallistatin and afamin in the blood serum of patients after a myocardial infarction and without a cardiovascular event, and determined the relationship between the concentration of kallistatin and afamin and the anthropometric indicators of being overweight and of obesity in these groups. Serum kallistatin and afamin were quantified by ELISA tests in a cross-sectional study of 160 patients who were divided into two groups: study group (SG) (*n* = 80) and another with no cardiovascular event (CG) (*n* = 80). Serum kallistatin concentration was significantly higher in the SG (*p* < 0.001), while the level of afamin was significantly lower in this group (*p* < 0.001). In addition, a positive correlation was observed in the SG between the afamin concentration and the waist to hip ratio (WHR), lipid accumulation product (LAP) and the triglyceride glucose index (TyG index). In the CG, the concentration of kallistatin positively correlated with the LAP and TyG index, while the concentration of afamin positively correlated with all the examined parameters: body mass index (BMI), waist circumference (WC), hip circumference (HC), waist to hip ratio (WHtR), visceral adiposity index (VAI), LAP and TyG index. Serum kallistatin and afamin concentrations are associated with the anthropometric parameters related to being overweight and to obesity, especially to those describing the visceral distribution of adipose tissue and metabolic disorders related to excessive fatness.

## 1. Introduction

Obesity is one of the major risk factors for cardiovascular disease (CVD) and many other chronic diseases, such as type 2 diabetes, dyslipidemia and hypertension [1]. Several reports have shown that adipose tissue (AT) contains different types of cells [2]. In obesity, hypertrophic AT shows abnormal metabolic properties, contributing to CVD development in obese patients [3]. However, some adipokines have anti-inflammatory properties, preventing adipocyte hypertrophy and dysfunction [4]. Anti-inflammatory adipokines, such as adiponectin, are potent in counteracting inflammation, hyperglycemia, or lipotoxic damage [5]. An endogenous plasma protein called kallistatin demonstrates a similar pleiotropic effect, such as the inhibition of inflammation, oxidative stress, fibrosis, apoptosis, and angiogenesis.

Kallistatin, also known as serpin family A member 4, was first discovered in human serum as a tissue-kallikrein-binding protein and a unique serine proteinase inhibitor. The inhibitory effect of kallistatin on tissue kallikrein is blocked by its interaction with heparin [6]. As a pleiotropic protein, kallistatin has several biological functions, including being an anti-inflammatory, antioxidant, anti-angiogenesis, and vasodilator [7]. Some reports suggest that the overexpression of kallistatin improves myocardial ischemia-reperfusion damage, inhibits heart remodeling after a myocardial infarction (MI), and alleviates hypertension in animal models, suggesting a close relationship between kallistatin and CVD [8]. Additionally, it is worth noting that scientific evidence indicates that kallistatinis able to inhibit the development of atherosclerosis, which may indicate a new therapeutic target for diseases related to atherosclerosis [9].

Human afamin is a glycoprotein that binds vitamin E (α-tocopherol) in human plasma. Blood afamin levels are associated with many diseases, such as obesity, pregnancy pathologies, polycystic ovary syndrome, type 2 diabetes, hypertension, and dyslipidemia [10]. In addition, this protein plays an important role in the anti-apoptotic cellular processes related to oxidative stress, and isassociated with insulin resistance and other components of metabolic syndrome [11]. Other experimental studies, conducted on transgenic mice that were overexpressing the human afamin gene, showed that these experimental animals increased their body weight, lipid concentration and blood glucose levels [12]. In clinical trials, Polkowska et al. [13] showed a positive correlation between the level of afamin and the body mass index (BMI) in the group of children with type 1 diabetes.

Despite intensive research, the body of knowledge concerning kallistatin and afamin’s physiological and pathological properties in obese and myocardial infarction patients is still limited. Most studies assessing the relationship of kallistatin and afamin with being overweight and obese have been conducted in people with type 1 or 2 diabetes and animal models. No studies describe this relationship in patients after myocardial infarction hospitalized in the initial phase of a cardiac rehabilitation program, continuing exercise, dietary and pharmacological therapy. Moreover, our knowledge of the importance of regulatory peptides during cardiac convalescence is still insufficient, and their level in blood serum may become an important therapeutic target in the secondary prevention of CVD, especially in overweight or obese people. Therefore, the study aimed to assess the concentration of kallistatin and afamin in the blood serum of patients after myocardial infarction and without a cardiovascular event, and to determine the relationship between the concentration of kallistatin and afamin and the anthropometric indicators of being overweight and obese in these groups.

## 2. Materials and Methods

The study was conducted in accordance with the Helsinki Declaration (updated in 2013). The study was approved by the Bioethics Committee of the Medical University of Lublin (Lublin, Poland) (KE-0254/197/2017), and all respondents participating in the study gave their informed consent in writing. This cross-sectional study was conducted from August to December 2017 among 160 adults divided into two groups: the study group (SG) and the control group (CG).

### 2.1. Study Population

The study group (SG) consisted of patients post-myocardial infarction, hospitalized in the early period of cardiac rehabilitation (up to 14 days after discharge from the end of complete revascularization), continuing physical therapy, dietary and pharmacological therapy in two health and spa facilities in eastern Poland. The study included patients who met the inclusion criteria and provided informed consent to participate from subsequent convalescence stay. All patients in this group had had their first myocardial infarction and were treated with primary percutaneous coronary intervention (PCI). The inclusion criteria for the study group included: age 40–65 years, state after myocardial infarction, and consent to participate in the study in writing. Exclusion criteria included renal failure, cancer, history of lung disease or rheumatic disease, age under 40 and over 65. Other exclusion criteria included factors affecting serum kallistatin and afamin levels, such as dietary supplements, vitamin complexes, and alcohol dependence.

The control group (CG) consisted of adults without a cardiovascular incident in the past, presenting for control and prophylactic examinations to an occupational medicine physician as part of periodic examinations. The respondents were recruited in an occupational medicine center in eastern Poland. Inclusion criteria for the study included: age between 40 and 65 years, no history of a cardiovascular event, alow 10-year risk of a cardiovascular event (SCORE < 5) [14], no chronic diseases (hypertension, diabetes, renal failure, neoplastic, rheumatic and lung diseases), no cardiovascular complaints that could suggest atherosclerotic CVD, no dietary supplements, pre-diabetes, orhypercholesterolemia. Exclusion criteria included active infection and a history of alcoholism.

### 2.2. Blood Sampling and Laboratory Tests

Blood samples were taken from the ulnar vein in the morning on an empty stomach (7:00–9:00) after an overnight rest into a test tube with a clotting activator and a separating agent (granules), and delivered to the laboratory within 1 h. Until the blood was delivered to the laboratory, the samples were stored at 4 °C and protected from daylight. Plasma was separated by centrifugation at 3000 rpm for 10 min. After centrifugation, the sera were transferred to Eppendorftest tubes, immediately frozen, and stored at −80°C until analysis.

The serum lipid profile (total cholesterol (TC), triglycerides (TG), HDL cholesterol (HDL-C)), glucose (FBG) and creatinine levels were determined using standard laboratory methods. LDL cholesterol was calculated from the Friedewald formula (when the concentration of TG < 400 mg/dL), eGFR (estimated glomerular filtration rate) from the Cockcroft-Gault formula, and non-HDL using the non-HDL = TC [mg/dL] formula—HDL-C [mg/dL] [15]. The quantitative enzyme immunoassay was used to measure the concentration of kallistatin and afamin, and the determinations were made by ELISA using the original reagents (ImmunDiagnostik, Bensheim, Germany).

### 2.3. Anthropometric Measurements

All subjects were subjected to anthropometric measurements of height and weight. First, height was measured with an accuracy of 0.1 cm with an altimeter, and body weight was measured without shoes and outerwear with a platform scale with an accuracy of 0.1 kg. Then, the BMI was calculated for all subjects, defined as body weight in kilograms (kg) divided by height in meters squared (kg/m^2^) [16].

A non-elastic tape measure was used to measure the waist circumference (WC)between the lower edge of the costal arch and the upper crest of the ilium, and the circumference of the hip (HC)at the height of the greater trochanter of the femur. During both measurements, the participant exhaled and his legs were open about 25–30 cm to distribute the body weight. Both measurements were made in a standing position. Then, the waist circumference to hip circumference ratio (WHR) and the waist circumference to height ratio (WHtR) were calculated [17].

The electrical bioimpedance method was used to assess the percentage of adipose tissue (FM%) using a body composition analyzer (OMRON Model BF306) according to the algorithm developed by the manufacturer.

Based on anthropometric measurements and biochemical results, the respondents had calculated the visceral adiposity index (VAI) and the body adiposity index (BAI). The VAI used the equation [WC/(36.58 + (1.89 × BMI))] × (TG × 0.81) × (1.52/HDL) for women and [WC/(39.68 + (1.88 × BMI))] × (TG/1.03) × (1.31/HDL) for men [17]. The BAI was calculated using the equation [HC (cm)/height (m) 1.5] − 18 [18]. Lipid accumulation product (LAP) was calculated as [(WC-65) × TG level] for men and [(WC-58) × TG level] for women [19]. TG and glucose index (TyG index) was calculated from the Ln equation [(TG level [mg/dL] × fasting glucose level [mg/dL]/2)] [20].

### 2.4. Other Variables

The interviews with the respondents and the physical examination were performed by nurses specially trained for the study purpose. Sociodemographic data (age, sex, place of residence, education) was obtained using standard questionnaires. A smoker was defined as a respondent who smoked at least one cigarette a month [21].

### 2.5. Statistics Analysis

Continuous variables were presented as means with standard deviation (SD) if normally distributed, or median with first quartile (Q1) and third quartile (Q3). The Kolmogorov–Smirnov test was used to assess conformity with a normal distribution. Categorical variables were reported as absolute numbers and percentages. Differences between groups were assessed with the chi-square test for categorical variables and the Mann–Whitney U and *t*-test for continuous variables. Pearson correlations were used to investigate the relationships between the level of kallistatin, afamin and anthropometrics’ measurements. Moreover, linear regression was used to consider the influence of age, sex and smoking on the investigated relationships. The analysis of results is presented as a standardized beta coefficient with standard errors (SE). Due to skewed distribution, the logarithm transformation was applied. Statistical analyses were conducted using IBM Corp. Released 2017. IBM SPSS Statistics for Windows, Version 25.0. Armonk, NY, USA: IBM Corp. Statistical and Statistica 13 PL (StatSoft, Tulsa, OK, USA). A two-sided *p* < 0.05 was considered significant.

## 3. Results

### 3.1. Baseline Characteristics of the Participants

Table 1 presents the characteristics of the studied groups. The mean age in SG was 53.34 ± 4.74 and in CG 49.25 ± 6.23. There were no significant differences in the studied groups regarding sex, marital status, and family history of parental CVD (*p* > 0.05). Considering the anthropometric parameters describing the occurrence of overweight and obesity, SG was characterized by significantly higher values of BMI, WC, WHR, WHtR, VAI, LAP and TyG index compared to those observed in CG. The results of the biochemical tests are presented in the Appendix A.

### 3.2. Kallistatin and Afamin Levels among the Study Groups

Figure 1 illustrates the concentration of kallistatin and afamin in the study groups. The level of kallistatin was significantly higher in the studied group of patients after MI (*p* < 0.001). At the same time, the level of afamin was significantly lower in this group (*p* < 0.001) compared to those without a cardiovascular event.

### 3.3. Correlations between the Concentration of Kallistatin and Afamin and Anthropometric Measurements

Table 2 displays the relationship between the concentration of kallistatin, afamin and anthropometric measurements in the studied groups. A positive correlation was observed in SG between the afamin concentration and the WHR, LAP and TyG index. In CG, the concentration of kallistatin positively correlated with the waist circumference, WHR, LAP and TyG index, while the concentration of afamin positively correlated with all the examined parameters: BMI, WC, HC, WHtR, VAI, LAP and TyG index. The strongest relationships between afamin concentration and anthropometric measurements occurred in the case of FM (*r* = 0.60), BMI and TyG index (*r* = 0.54), and LAP (*r* = 0.46) in the control group.

### 3.4. Relationship between the Concentration of Kallistatin and Afamin and Anthropometric Measurements—Multivariate Models

Table 3 shows the relationship between kallistatin and afamin levels and anthropometric measurements after taking into account confounding variables (gender, age and smoking). In SG, the relationship between afamin concentration and WHR, LAP and TyG maintained the statistical significance, direction and strength of the relationship. In CG, after taking into account confounding variables, all significant variables (except WHR) in univariate models significantly related to afamin concentration maintained their statistical significance in multivariate models. In the case of kallistatin concentration, only the TyG index maintained statistical significance in multivariate models.

## 4. Discussion

Increasing knowledge regarding regulatory peptides allows for an assessment of their role in various non-communicable diseases, including CVD [13]. This cross-sectional study compared the concentration of kallistatin and afamin in patients with post-myocardial infarction and without a cardiovascular event, and assessed the relationship of kallistatin and afamin with the anthropometric indicators concerning being overweight and obese in these groups. In summary, the results of our study indicate that in the group of patients after myocardial infarction, significantly higher levels of kallistatin were observed, with simultaneously significantly lower levels of afamin. The second observation, resulting from the conducted research, concerns the relationship between the anthropometric parameters related to being overweight and obesity with the level of kallistatin and afamin. In the studied group of patients after myocardial infarction, a positive relationship of afamin was observed for the WHR, LAP and TyG index, while in patients without a cardiovascular event, it was observed that afamin positively correlated with all the assessed anthropometric parameters, except for WHR and percentage of adipose tissue.Moreover, in this group of patients, kallistatin positively correlated with the LAP and TyG index. After considering the confounding variables, the relationship and strength of the relationships were maintained in multivariate models, except for the WHR in CG.

Yao et al. [9] revealed that plasma kallistatin concentration in patients with coronary artery disease is significantly reduced and shows a negative correlation with the severity of its symptoms. Similarly, Zhang et al. [22] found decreased plasma kallistatin levels in patients with stable angina and acute coronary syndrome compared to healthy controls. However, in those two studies, patients’ blood samples were collected prior to coronary angiography, in contrast to our studies where we collected blood samples after PCI during cardiac rehabilitation. Sirchak et al. [23] concluded that patients with pancreatitis and atherosclerosis have lower levels of circulating kallistatin than patients with pancreatitis alone. In our study, serum kallistatin concentration was found to be significantly higher in patients after myocardial infarction than in the control group. According to our knowledge, this is the first study conducted on a group of patients after a myocardial infarction during cardiac rehabilitation. One of the possible mechanisms explaining the obtained result is that the increased level of kallistatin in patients after myocardial infarction may perform a repair function after a cardiovascular incident. As shown in studies on rats, kallistatin increases in vitro growth, proliferation, and the migration of vascular smooth muscle cells, and inhibits the in vitro proliferation, migration, and adhesion of vascular endothelial cells. The Miao et al. [24] observationsindicate that kallistatin is elevated at the level of transcription in damaged vessels, suggesting that kallistatin may play an essential role in autocrine growth factors in mediating proliferation and migration in aortic vascular smooth muscle cells (VSMCs) and in neovascularization after balloon angioplasty. Similarly, Chao et al. [7] and Gao et al. [8] observed that overexpression of kallistatin improves myocardial ischemia-reperfusion damage by inhibiting heart remodeling after myocardial infarction in animal models. Thus, our result may indicate the ongoing intensive repair processes in the endothelium of damaged coronary vessels after myocardial infarction and PCI performed in the studied group of patients. Nevertheless, more research is needed to establish the role of kallistatin in the recovery process following myocardial infarction.

Kallistatin was initially identified as an important factor in inhibiting inflammation, oxidative stress, fibrosis, and angiogenesis [25]. However, the role of kallistatin as an element associated with obesity has not yet been fully described. Zhu et al. [26] demonstrated an inverse association of kallistatin with obesity in a group of apparently healthy African-American adolescents. By contrast, Gateva et al. [27] found no correlation between the level of kallistatin and BMI, WHR, waist to height ratio (WSR) and VAI in obese patients without glycemic disorders and in the group of obese patients with pre-diabetes. Frühbeck et al. [28] assessed the concentration of kallistatin and its association with obesity in healthy people and the group of obese patients. They observed that the level of circulating kallistatin was significantly reduced in the obese group compared to the lean group. Moreover, they observed a negative relationship between circulating kallistatin levels and obesity-related anthropometric parameters. Interestingly, in the one-year follow-up of the cited studies, it was found that the decreased level of kallistatin in obese subjects increased after weight loss. The authors’ research showed no positive relationship between anthropometric parameters and the level of circulating kallistatin in the study group. In contrast, a positive relationship was found between the LAP and TyG index and circulating kallistatin in the control group. However, popular anthropometric parameters such as BMI, WC, WHR, WHR, and WHtR provide limited information on adipose tissue distribution in the body. They do not reflect the distribution of visceral adipose tissue.Therefore, other parameters have been developed that take into account the accumulation and distribution of adipose tissue, such as LAP, which is an indicator of excessive abdominal fat accumulation based on triglyceride and WC levels [19]. The results of our research may suggest that kallistatin may play a protective role against the excess of visceral fat, which is strongly associated with metabolic and obesity-related disorders. However, this thesis should be confirmed in other studies because, in both studied groups, many risk factors may be related to the level of circulating kallistatin.

There are few studies assessing afamin concentration in the context of CVD. In a pilot study, Ward et al. [29] assessed differences in proteomic profiles of atherosclerotic lesions (samples were collected in three different regions of atherosclerotic tissue) between female and male patients. They observed that atherosclerotic lesions were more abundant in afamin and serine protease inhibitors in women in the three sampled areas of the atherosclerotic lesion, suggesting that women develop atherosclerotic plaque with a lower inflammatory profile and greater stability than men. The authors’ research showed that plasma afamin concentration was significantly lower in patients after myocardial infarction than the study participants without a cardiovascular event. Similarly, Hasan et al. [30] observed a significantly higher concentration of afamin in the control group compared to the study group of patients with myocardial infarction before coronary catheterization. Interestingly, in the same study, it was observed that afamin concentration was significantly higher after coronary catheterization than before. A possible explanation could be the different degrees of inflammation found in patients in the acute and chronic phases of the disease.

Several studies showed a positive relationship between afamin concentration and obesity, e.g., in obese children [31], and children with diabetes [13].Oller Moreno et al. made interesting observations [32] evaluating the plasma proteome profile in a cohort of people who underwent nutritional intervention for weight loss. They observed a decrease in afamin concentration with weight loss. In a large-scale population study, it was found that each increase in afamin concentration by 10 mg/L resulted in a 19% increase in the components of the metabolic syndrome [12]. The authors’ research showed that afamin concentration positively correlated with anthropometric parameters describing overweight and obesity. The strongest correlation was observed with the FM, BMI and TyG index in CG. However, more research is needed to elucidate the effects of afamin and its relationship to adipose tissue in CVD subjects, which could lead to the identification of new roles of this marker in the search for new therapeutic agents for the CVD epidemic.

### Strengths and Limitations

An extensive analysis of the literature has shown that few studies are assessing the relationship between kallistatin and afamin and obesity-related anthropometric indices in the group of patients after myocardial infarction. Second, an analysis of the available literature showed that, in the studies described so far, blood samples for the determination of kallistatin and afamin levels in patients with CAD were collected during the chronic phase of the disease or at the time of exacerbation of symptoms prior to PCI. Therefore, we conducted our study among patients during cardiac rehabilitation in the period of the body’s ongoing repair processes related to past MI.

Nevertheless, this study has several limitations. First, it is a cross-sectional study and thus shows no cause–effect or time–effect relationship between kallistatin and afamin levels and obesity-related anthropometric parameters. Secondly, only basic sociodemographic and lifestyle data (i.e., gender, age, and smoking) were included in the models analyzed. Second, we included a relatively small number of patients per center, especially for SG. The third limitation of our study is the use of electrical bioimpedance analysis (BIA), which is an indirect method of assessing body composition; however, comparative studies have shown a significant correlation between data obtained from BIA measurements and body composition measured by densitometry, which is the gold standard for this type of study; however, it is associated with exposure to X-rays [33]. The fourth limitation of our study was not including medications taken by patients in particular groups; although the main objective of the study was to evaluate parameters related to obesity and the concentration of kallistatin and afamin, medications could have influenced the results of the study.Future studies should investigate whether the levels of kallistatin and afamin affect the speed of recovery in patients after myocardial infarction.

## 5. Conclusions

In conclusion, in this study, higher concentrations of kallistatin and lower levels of afamin were observed in the group of patients after myocardial infarction, after PCI, during cardiac rehabilitation. Moreover, the concentration of kallistatin and afamin positively correlates with obesity-related anthropometric measures, especially those describing the excessive accumulation of abdominal fat (LAP) and obesity-related metabolic disorders (TyG index). Given the significant relationship between kallistatin and afamin levels and obesity-related anthropometric parameters, further studies in a larger group are required to investigate the relationships, taking into account the intermediary variables in these relationships, e.g., medications that the patient is taking. Nevertheless, conclusions from this study suggest that kallistatin and afamin, in combination with anthropometric indicators related to overweight and obesity, may play an important role duringrecovery in MI patients, and that the levels of these two regulatory proteins may become a therapeutic target. Therefore, after MI, overweight or obese patients are postulated to up-regulate kallistatin levels as it may be an important regulator in protecting post-MI injury and cardiac re-modeling. On the other hand, in the case of afamin, it is postulated to down-regulate its level because too high a level may result in excessive weight gain and thus an increase in the components of the metabolic syndrome, which is also associated with an increase in the number of risk factors for CVD, which is important in secondary prevention of CVC.

## Figures and Tables

**Figure 1 jcm-10-05792-f001:**
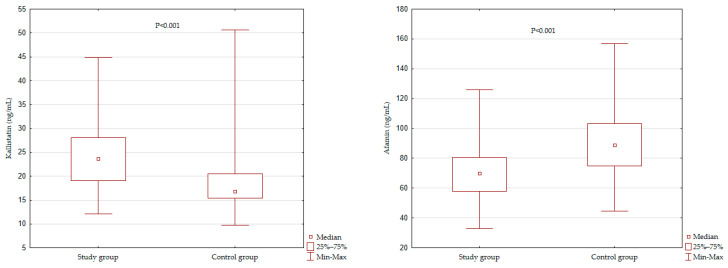
Kallistatin and afamin serum level comparison of studied groups.

**Table 1 jcm-10-05792-t001:** Baseline demographics and clinical characteristics.

Variables	Study Group (*n* = 80)	Control Group (*n* = 80)	*p*
Demographic and Clinical Data:			
Age (years)	53.34 ± 4.74	49.25 ± 6.23	<0.001
Men	22 (27.5)	28 (35)	0.31
In Relationship	64 (80)	69 (86.3)	0.398
Current Smoking	25 (31.3)	9 (11.3)	<0.001
Family History of CVD—on the Mother’s Side	46 (57.5)	42 (52.5)	0.634
Family History of CVD—on the Father’s Side	49 (61.3)	39 (48.8)	0.153
Diabetes	19 (24)	0 (0)	<0.001
Arterial Hypertensions	54 (68)	0 (0)	<0.001
Anthropometric Variables:			
BMI (kg/m^2^)	28.89 ± 4.91	26.64 ± 4.04	0.002
WC (cm)	103.9 ± 12,48	93.36 ± 12.50	<0.001
HC (cm)	105 ± 10,83	102.9 ± 7.72	0.15
WHR	0.99 ± 0.08	0.91 ± 0.09	<0.001
WHtR	0.60 ± 0.07	0.54 ± 0.07	<0.001
FM (%)	31.08 ±7.71	29.65 ± 7.05	0.22
VAI	2.12 ± 1.56	1.06 ± 0.7	<0.001
BAI	28.63 ± 6.44	27.79 ± 4.73	0.35
LAP	60.9 (43.4–89.6)	48.3 (21.8–60)	<0.001
TyGindex	4.8 (0.26)	4.7 (0.27)	0.001

Note: Continuous data are presented as: means ± standard deviation (SD) or median (interquartile range) and categorical data was shown as *n* (%); *p*-value in this table was analyzed between two groups. Abbreviations: CVD, cardiovascular disease; BMI, body mass index; WC, waist circumference; HC, hip circumference; WHR, waist to hip ratio; WHtR, waist-to-height ratio; FM%, body fat percentage; VAI, visceral adiposity index; BAI, body adiposity index; LAP, lipid accumulation product; TyG index, the triglyceride glucose index.

**Table 2 jcm-10-05792-t002:** Correlation between kallistatin and afamin serum level and anthropometric measures.

Variables	Log Kallistatin (ng/mL)		Log Afamin (ng/mL)	
	*r*	*p*	*r*	*p*
Study Group:				
BMI	−0.156	0.17	0.13	0.231
WC	−0.12	0.27	0.14	0.210
HC	−0.12	0.31	−0.03	0.812
WHR	−0.05	0.67	0.25	0.026
WHtR	−0.12	0.30	0.12	0.295
FM	0.02	0.89	0.09	0.41
VAI	0.13	0.23	0.20	0.078
BAI	−0.08	0.48	−0.04	0.742
LAP	0.11	0.31	0.28	0.013
TyG index	0.11	0.33	0.35	0.002
Control Group:				
BMI	0.15	0.19	0.54	<0.001
WC	0.25	0.02	0.47	<0.01
HC	0.12	0.30	0.38	0.001
WHR	0.25	0.03	0.34	0.002
WHtR	0.18	0.11	0.10	<0.001
FM%	−0.01	0.92	0.60	<0.001
VAI	0.17	0.13	0.29	0.008
BAI	−0.09	0.45	0.39	0.002
LAP	0.23	0.04	0.46	<0.001
TyG index	0.33	<0.001	0.54	<0.001

Abbreviations: r, correlation coefficient; BMI, body mass index; WC, waist circumference; HC, hip circumference; WHR, waist to hip ratio; WHtR, waist-to-height ratio; FM%, body fat percentage; VAI, visceral adiposity index; BAI, body adiposity index; LAP, lipid accumulation product; TyG index, the triglyceride glucose index.

**Table 3 jcm-10-05792-t003:** Relationship between the concentration of kallistatin and afamin and anthropometric parameters describing overweight and obesity in multivariate models.

Variables	Log Kallistatin (ng/mL)				Log Afamin (ng/mL)			
	Model A		Model B		Model A		Model B	
	b (SE)	*p*	b (SE)	*p*	b (SE)	*p*	b (SE)	*p*
Study Group:								
BMI	−0.16 (0.11)	0.164	−0.16 (0.11)	0.165	0.13 (0.11)	0.268	0.13 (0.11)	0.249
WC	−0.14 (0.12)	0.240	−0.14 (0.12)	0.240	0.14 (0.12)	0.230	0.15 (0.12)	0.206
HC	−0.13 (0.12)	0.289	−0.13 (0.12)	0.288	−0.05 (0.12)	0.674	−0.04 (0.12)	0.727
WHR	−0.08 (0.14)	0.555	−0.08 (0.14)	0.557	0.36 (0.13)	0.006	0.37 (0.13)	0.006
WHtR	−0.12 (0.12)	0.294	−0.12 (0.12)	0.295	0.11 (0.12)	0.352	0.11 (0.12)	0.273
FM%	0.05 (0.16)	0.746	0.05 (0.16)	0.750	0.16 (0.16)	0.326	0.17 (0.16)	0.300
VAI	0.14 (0.11)	0.237	0.16 (0.12)	0.195	0.20 (0.11)	0.083	0.19 (0.12)	0.131
BAI	−0.12 (0.14)	0.410	−0.12 (0.15)	0.410	−0.10 (0.15)	0.507	−0.09 (0.15)	0.539
LAP	0.11 (0.11)	0.323	0.13 (0.12)	0.300	0.28 (0.11)	0.013	0.27 (0.11)	0.02
TyG index	0.11 (0.11)	0.346	0.13 (0.12)	0.31	0.36 (0.11)	0.001	0.36 (0.12)	0.002
Control Group:								
BMI	0.08(0.11)	0.499	0.07(0.11)	0.535	0.48(0.10)	<0.001	0.48(0.10)	<0.001
WC	0.14(0.12)	0.267	0.13(0.13)	0.291	0.39(0.11)	0.001	0.39(0.12)	0.001
HC	0.11(0.11)	0.330	0.10(0.11)	0.345	0.38(0.10)	0.000	0.38(0.10)	<0.001
WHR	0.11(0.14)	0.421	0.11(0.14)	0.457	0.19(0.14)	0.164	0.20(0.14)	0.167
WHtR	0.13(0.11)	0.240	0.13(0.11)	0.265	0.33(0.10)	0.002	0.33(0.11)	0.002
FM%	0.31(0.14)	0.024	0.31(0.14)	0.028	0.60(0.12)	<0.001	0.63(0.12)	<0.001
VAI	0.10(0.11)	0.349	0.10(0.11)	0.358	0.23(0.11)	0.031	0.23(0.11)	0.032
BAI	0.15(0.13)	0.262	0.15(0.13)	0.282	0.39(0.12)	0.003	0.39(0.13)	0.003
LAP	0.16(0.11)	0.151	0.16(0.11)	0.160	0.40(0.10)	<0.001	0.40(0.10)	<0.001
TyG index	0.25(0.11)	0.027	0.48(0.10)	<0.001	0.48(0.10)	<0.001	0.50(0.10)	<0.001

Model A: adjusted for age and sex; Model B: adjusted for age, sex and smoking status. Abbreviations: b, standardized beta coefficient; SE: standard error; BMI, body mass index; WC, waist circumference; HC, hip circumference; WHR, waist to hip ratio; WHtR, waist-to-height ratio; FM%, body fat percentage; VAI, visceral adiposity index; BAI, body adiposity index; LAP, lipid accumulation product; TyG index, the triglyceride glucose index.

## Data Availability

The datasets used and/or analyzed during the current study are available from the corresponding author on reasonable request.

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
