# Peer review of "Relationship between Serum Kallistatin and Afamin and Anthropometric Factors Associated with Obesity and of Being Overweight in Patients after Myocardial Infarction and without Myocardial Infarction"

_jcm, 2021, doi:10.3390/jcm10245792_

Round 1

Reviewer 1 Report

In this study entitled “Relationship Between Serum Kallistatin and Afamin and An-2 thropometric Factors Associated with Obesity and Overweight 3 in Patients After Myocardial Infarction and Without Myocar-4 dial Infarction”, the authors investigated the association between kallistatin/afamin and obesity-related parameters in patients following myocardial infarction. They found that the levels of kallistatin were higher and those of afamin were lower than the control group following myocardial infarction. These parameters were correlated with obesity-related anthropometric measurements. Several concerns have been raised.

  1. The manuscript is overall too long and redundant. Of note, the introduction and discussion section are too long. Please attempt to more summarize then. Of note, the numbers of reference is too much (>80) as if this is a review article. In general, the reference numbers would be around 30 for the original article.

  1. Could the authors explain a little bit more the reason why they focused on the patients following myocardial infarction? It is stated that there are no studies that investigated such a cohort, but this explanation would be insufficient.

  1. Could the authors explain the clinical implication of this finding in the daily clinical practice for the clinicians? In the conclusion section, the authors state that these parameters would be risk stratify the patients. However, it seems that there is no analysis for the risk assessments.

  1. Some patients might have taken medications that have considerable impacts on obesity-related parameters.

Reviewer 2 Report

In this manuscript, Nowicki and colleagues measured serum kallistatin and afamin levels in patients after myocardial infarction (n=80) and the control group with no cardiovascular event (n =80) using ELISA based assay. Since obesity has been linked to cardiovascular diseases, the authors also considered anthropometric factors while analyzing the data. In these patients, kallistatin was not corrected with anthropometric factors, and afamin only showed a positive correlation with WHR, LAP, and TyG index. In the control group, the concentration of kallistatin positively correlated with the LAP and TyG index, and the concentration of afamin positively correlated with all of the measured anthropometric factors. Interestingly, the authors observed that serum kallistatin concentration was significantly higher and afamin was significantly lower in the recovery stage after myocardial infarction of patients, indicating unique roles of serum kallistatin and afamin in the recovery stage after myocardial infarction. The reviewer only has a few suggestions:

  1. The sentence: “The obtained results…” in lines 25-27 seems redundant with the sentence in lines 32-35. Can the authors modify accordingly?
  2. The Figure of “Kallistatin and afamin serum level comparison of studied groups” should be Figure 1.
  3. In lines 211-213, it will be more convenient for reading, if the authors can indicate that “The strongest relationships between afamin concentration and anthropometric measurements occurred” is in the control group.
  4. Do the patients with higher kallistatin and low afamin recover faster/better from myocardial infarction?

Round 2

Reviewer 1 Report

Reference number is still too much (>50). Introduction and discussion sections were further increased. 

Author Response

Respected Reviewer,

Thank you very much for your review. We appreciate your suggestions and comments, which will undoubtedly improve the quality of our manuscript.

We wish to submit a revised manuscript “Relationship Between Serum Kallistatin and Afamin and Anthropometric Factors Associated with Obesity and Overweight in Patients After Myocardial Infarction and Without Myocardial Infarction”(Manuscript ID: jcm-1490014) for further consideration by the Journal of Clinical Medicine.

We do confirm that we have taken all the valuable reviewers comments under consideration and applied changes to the manuscript. The improvements are marked in blue in the manuscript.

Point 1: Reference number is still too much (>50). Introduction and discussion sections were further increased.

Response: Respected Reviewer, Thank you very much for your suggestion. We have furtherly reduced the number of references to 33 positions (from 55 in previously revised version). We have considerably shortened the discussion section (from 1943 words in the previously reviewed version to 1216 words in the current one). Similarly, the introduction has been visibly condensed (700 words to just above 500 words).

All the authors are unanimous in term of accepting above-mentioned changes. We hope that the version submitted will meet the criteria to be issued in the Journal of Clinical Medicine.

Thank you

Reviewer 2 Report

My questions/concerns have been fully addressed.

Round 3

Reviewer 1 Report

There are no further comments.